# The epidemiology, treatment patterns, healthcare utilizations and costs of Acute Myeloid Leukaemia (AML) in Taiwan

Huai-Hsuan Huang[1], Chieh-Min Chen[2], Chen-Yu Wang[3,4,5], William Wei-Yuan Hsu[6], Ho-Min Chen[7], Bor-Sheng Ko[1,8]*, Fei-Yuan Hsiao[2,3,9]*

1 Division of Hematology, Department of Internal Medicine, National Taiwan University Hospital, Taipei, Taiwan, 2 Graduate Institute of Clinical Pharmacy, College of Medicine, National Taiwan University, Taipei, Taiwan, 3 School of Pharmacy, College of Medicine, National Taiwan University, Taipei, Taiwan, 4 Department of Pharmacy, National Taiwan University Hospital Yun-Lin Branch, Yun-Lin County, Taiwan, 5 Preparatory Office of National Center for Geriatrics and Welfare Research, Yun-Lin County, Taiwan, 6 Department of Computer Science and Engineering, National Taiwan Ocean University, Keelung, Taiwan, 7 Health Data Research Center, National Taiwan University, Taipei, Taiwan, 8 Department of Hematological Oncology, National Taiwan University Cancer Center, Taipei, Taiwan, 9 Department of Pharmacy, National Taiwan University Hospital, Taipei, Taiwan

☯ These authors contributed equally to this work.
* fyshsiao@ntu.edu.tw (FYH); bskomd@ntu.edu.tw (BSK)

**Data Availability Statement:** All relevant data are within the paper and its Supporting Information files.

## Abstract

### Backgrounds

An increasing incidence of Acute Myeloid Leukaemia (AML) has been reported in several Western countries. However, the epidemiology of AML in Asia is very limited. According to the National Comprehensive Cancer Network (NCCN) guideline of AML, a range of conventional therapy options is available to AML patients. Nevertheless, different treatment strategies may result in diverse healthcare utilization and costs. Understanding the treatment patterns, healthcare utilization and costs of AML would thus be essential for clinicians and policymakers to optimize the treatment strategies of AML.

### Objectives

The objective of this study was to investigate the incidence, treatment patterns, healthcare utilization and costs of AML in Taiwan using a nationwide population database.

### Methods

We retrospectively identified AML patients diagnosed from 2006 to 2015 from the Taiwan Cancer Registry Database (TCRD) and estimated the epidemiology of AML in Taiwan. The TCRD was linked to National Health Insurance Research Database (NHIRD) to collect the treatment patterns and health care utilization. Patients diagnosed with AML from 2011 to 2015 were further identified to analyze treatment patterns, healthcare utilization and costs.

**Funding:** Huai-Hsuan Huang, Chieh-Min Chen, Chen-Yu Wang, Ho-Min Chen, Bor-Sheng Ko and Fei-Yuan Hsiao received grants from AbbVie Biopharmaceuticals and grants from Ministry of Science and Technology (MOST), Taiwan (MOST 108-2314-B-002-118-MY3), during the conduct of the study.

**Competing interests:** The authors have declared that no competing interests exist.

## Results

The crude annual incidence of AML increased from 2.78 to 3.21 cases per 100,000 individuals from 2006 to 2015. However, the age-standardized rate (ASRs) of AML slightly declined from 2.47 to 2.41 cases per 100,000 individuals in the same period. Among 2,179 AML patients who received induction therapy (median age: 56 years), most of them (n = 1744; 80.04%) received standard-dose cytarabine (SDAC) regimen. The remaining 162 patients received high dose cytarabine (HDAC) and 273 patients received non-standard dose cytarabine (N-SDAC) regimen as the induction therapy. The median medical costs in our study for patients treated with chemotherapy alone was $42,271 for HDAC, $36,199 for SDAC and $36,250 for N-SDAC. For those who received hematopoietic stem cell transplantation (HSCT) after induction therapy, their median medical costs were $78,876 for HDAC, $78,593 for SDAC and $79,776 for N-SDAC.

## Conclusions

This study is the first population-based study conducted in Asia to provide updated and comprehensive information on epidemiology, treatment patterns and healthcare resource utilization and costs of AML.

## Introduction

Acute myeloid leukaemia (AML) is a rare heterogeneous disease comprising a group of hematopoietic neoplasms originating from the excessive clonal proliferation of myeloid precursor cells [1]. It is the most common acute leukaemia in adults. It accounts for 1.3% of new cancer patients in the United States, with an incidence rate of 4.3 cases per 100,000 individuals in 2018 [2]. In Europe, the incidence of AML has increased in years and the highest incidence was found in the UK, with an incidence rate of 4.05 cases per 100,000 individuals in 2017 [3,4]. AML occurs predominantly in the older population with a median age at diagnosis of 68 to 70 years [2–4]. As the population ages worldwide, the incidence of AML has increased globally. This circumstance increases the clinical and economic burden of AML.

According to the National Comprehensive Cancer Network (NCCN) and the European Society for Medical Oncology (ESMO) guidance [4], induction therapy of AML typically includes cytarabine as the backbone with other intravenous antineoplastic or immunomodulating agents [4]. AML patients may undergo allogeneic hematopoietic stem cell transplantation (HSCT) as consolidation therapy after the first complete remission from induction therapy [5]. However, treatment strategies differ by age, pre-treatment comorbidities, biological risk categories and patient preferences, which thus result in diverse healthcare utilization and costs [4,6]. Real-world data regarding epidemiology, treatment patterns, healthcare utilization and costs of AML would be essential for clinicians and policymakers to optimize the treatment strategies of AML. Notably, existing studies regarding this issue are conducted in Western countries [7] and there is a lack of these data in the Asian population.

The aim of this study was thus to provide the population-based epidemiology of AML in the Asia-pacific region. The study further aimed to fill the knowledge gaps regarding the current treatment patterns, healthcare utilization and costs of AML under Taiwan's National Health Insurance system.

## Materials and methods

### Data source

The study cohort was extracted from the Taiwan Cancer Registry Database (TCRD) and linked to Taiwan National Health Research Database (NHIRD) and National Death Registry (NDR) to conduct a population-based retrospective cohort study.

The TCRD is a nationwide database and the information of newly diagnosed cancer patients in hospitals with 50 or more bed capacity is reported to TCRD [8]. The diagnosis codes of cancer were based on the International Classification of Diseases for Oncology, Third Edition (ICD-O-3) [9]. The definitions of terminology, coding and procedures of reporting system were validated by the Taiwan National Public Health Association. Since 2003, the data completeness of TCRD has reached 98.4%; the percentage of morphological verification has increased from 87.1% to 93% [8,10].

The NHIRD is a nationwide claim-based database and covers over 99% of the Taiwan population, which can be used to estimate the epidemiology, prescription patterns and disease burden of all beneficiaries under National Health Insurance (NHI) in Taiwan [11]. The NHIRD includes all clinical information through the single-payer NHI programme, including patient demographic characteristics, ambulatory care records, admissions, procedures, diagnoses, prescribed medications and billing orders [12,13].

The NDR provides comprehensive information on the cause of death and the date of death. The identification numbers of all selected cancer beneficiaries from the databases mentioned above were linked by scrambled individual patients' confidentiality and analyzed under the Health and Welfare Data Science Centre (HWDC), Ministry of Health and Welfare, Taiwan.

### Ethical statement

The identification numbers of the beneficiaries were encrypted to ensure their confidentiality. However, unique identification numbers allowed for interconnections among all database subsets of the NHI program. The protocol of this study was approved by the Research Ethics Committee of National Taiwan University Hospital (Registration number, 201910016RINA).

### Study population

We enrolled all newly diagnosed AML patients between January 01 2006 and December 31 2015. The list of ICD-O-3 Code for AML was listed in **S1 Table**. For the treatment evaluation analysis, we only included patients diagnosed with AML between 2011 and 2015 because the detailed information of induction treatment status was only available from 2011 in the NHIRD. We further excluded patients less than 20 years old, without pathology confirmation and with acute promyelocytic leukaemia (APL) as they were considered to have a better prognosis and received different treatments. To analyze healthcare utilization and costs, we further excluded patients who did not receive induction therapy. All identified patients were further classified into six groups according to their treatment regimens and status of HSCT.

### Epidemiology of AML and patient characteristics

We estimated the annual incidence, age-standardized rate, age at diagnosis and sex ratio of newly diagnosed AML from 2006 to 2015. As the detailed information of induction treatment status was only available from 2011 in the NHIRD, we only analyzed baseline characteristics of patients with newly diagnosed AML and received induction therapy from 2011 to 2015.

## Treatment patterns of induction therapy

We evaluated the treatment patterns of patients who were diagnosed with AML from 2011 to 2015. Receiving induction therapy was defined as those who received an intravenous antineoplastic or an immunomodulating agent. Patients receiving induction therapy were categorized into three regimens based on the most updated NCCN guideline [14]. Due to the lack of information on the body surface area (BSA) in NHIRD and TCRD, we used an accumulated dose of cytarabine to define induction regimens. Patients who received less than 3000 mg of cytarabine within 21 days from the diagnosis date of AML were defined as receiving standard-dose cytarabine (SDAC) regimen. Those who received more than 3000 mg of cytarabine within 21 days from the diagnosis date of AML were defined as high-dose cytarabine (HDAC) regimens. All other patients who received medications apart from cytarabine, idarubicin and daunorubicin or received only idarubicin or daunorubicin within 21 days from the diagnosis date AML were defined as receiving a non-standard-dose cytarabine (N-SDAC) regimen. The age at induction therapy and time from diagnosis of AML to induction therapy were also analyzed.

## Definition of healthcare utilization and direct medical costs

We further evaluated healthcare utilization and costs of AML patients from 2011 to 2015 and followed up these patients for a maximum of 3 years. The follow-up period began on the date of the induction therapy of AML and ended on the date of death or December 31 2017, whichever came first. The healthcare utilization and associated direct medical costs were estimated annually for three consecutive years. The study time frame was shown in **S1 Fig**. Healthcare utilization and costs were presented for patients categorized by the induction regimen they received (SDAC, HDAC and N-SDAC regimen) and with or without HSCT. The number of outpatient department (OPD) visits, number of admissions, hospital length of stay (LOS) were assessed to calculate healthcare utilization. The calculation of the number of OPD visits and admissions per patient was divided by the number of patients who had OPD or admission claims recorded in the NHIRD. The LOS per patient is defined as dividing the length of stays by the number of hospitalized patients. Direct medical costs were calculated as all expenditures related to the treatment of AML, which included the costs of outpatient visits, hospitalizations, laboratory tests, prescription drugss, surgery, procedures (including radiation therapy or supportive care). The calculation of direct medical costs per patient involved dividing the total costs by the number of patients who survived during the follow-up period (the patient number survived was provided in the following tables and figures). All costs were reported as US dollars (the exchange rate was 30 New Taiwan dollars = 1 US dollar).

## Statistical analysis

The number of incident cases, age at diagnosis and sex ratio were calculated and reported annually for patients with AML from 2006 to 2015. The crude incidence rates were presented annually as cases per 100,000 individuals in the overall population of Taiwan and stratified by sex and age group. The age-standardized rates (ASRs), resented as cases per 100,000 individuals, were adjusted according to the World Health Organisation 2000 world standard population and reported annually [15]. For the treatment pattern of induction therapy, we reported the number of patients receiving treatment, age at treatment, time from diagnosis to induction therapy and types of induction regimen. Healthcare utilization and costs were stratified by regimens and with or without HSCT. Descriptive statistics were reported for continuous variables as median and interquartile ranges (IQRs) and for categorical variables as counts and percentages. All analyses used SAS version 9.4 (SAS Institute, Cary, NC, USA).

## Results

### Epidemiology of acute myeloid leukaemia; 2006 to 2015

From 2006 to 2015, 7,403 patients were enrolled and a slight increase trend of AML incident cases (636 in 2006 to 755 in 2015) was observed. (**Figs 1** and **2**). The crude annual incidence of AML increased from 2.78 to 3.21 cases per 100,000 individuals from 2006 to 2015, while the age-standardized rate (ASRs) of AML was slightly declined from 2.47 to 2.41 cases per 100,000 individuals in the same period (**Fig 2** and **S2 Table**). The sex ratio was decreased from 1.39 in 2006 to 1.2 in 2015. (**S2 Table**). The median age of the entire study population increased over the study period from 57 years in 2006 to 61 years in 2015. The AML incidence rates remained low in the age groups of 0–19 years and 20–39 years but increased in the age group of 40–59 years and reached a peak in the age group of more than 60 years. (**Fig 3** and **S2 Table**).

### Baseline characteristics for patients newly diagnosed with AML and received induction therapy from 2011 to 2015

We identified 3,292 adult patients with incident AML between 2011 and 2015 from the TCRD. Among them, 1113 (33.8%) patients who did not receive induction therapy were excluded (**Fig 1**). For those (n = 2179; 66.2%) who received induction therapy, their median age was 56 years with male predominant (**Table 1**). The median time from diagnosis to induction therapy was 7 days (IQR,3–18; **Table 1**). Most patients (79.1%) were treated in the medical center and 91.30% of patients was with Charlson comorbidity score (CCI) less than 2. (**Table 1**).

### Treatment patterns for patients newly diagnosed with AML and received induction therapy from 2011 to 2015

Among 2,179 AML patients, most of them (n = 1744, 80.04%) received SDAC regimen. Only 162 (7.43%) incident AML patients received HDAC and 273 (12.53%) patients received N-SDAC regimen as the induction therapy. The median age at treatment was 52 years (IQR,40–63; **Table 1**) for the HDAC group and 60 years (IQR,47–71; **Table 1**) for the N-SDAC group. The median time from diagnosis to induction therapy was 5 days (IQR,3–10; **Table 1**) and 13 days (IQR,4–37; **Table 1**) for HDAC and N-SDAC groups, respectively. For those receiving SDAC regimen as induction therapy, their median age at treatment was 56 years and the median time from diagnosis to induction therapy was 7 days (**Table 1**). Among these three groups, 361 (20.7%), 52 (32.1%) and 55 (20.1%) patients who received SDAC, HDAC and N-SDAC as induction therapy underwent HSCT respectively (**Fig 1**). Patients in the HDAC group received HSCT (n = 52, 32.1%) more often than patients in SDAC and N-SDAC groups, but their median time from induction therapy to HSCT was the longest. (214 days; IQR, 130–345 days; **Table 1**).

### Healthcare utilization and costs for patients newly diagnosed with AML and received induction therapy from 2011 to 2015

The median [IQR] duration of follow-up in the entire study population was 11.2 [2.4–32.4] months. During the first year, the median number of hospitalizations was the highest in HDAC patients with HSCT (5 [4–7]) followed by SDAC patients with HSCT (5 [4–6]), N-SDAC patients with HSCT (4 [3–6]), HDAC patients with chemotherapy alone (4 [1–5]), SDAC patients with chemotherapy alone (3 [1–5]) and N-SDAC patients with chemotherapy alone (3 [1–4]). The median length of stay, which represented the total amount of time in the first years after diagnosis that was spent in the hospital, was the longest in SDAC patients with HSCT (155 days [123–193]) and HDAC patients with HSCT (154 days [127–208]),. During

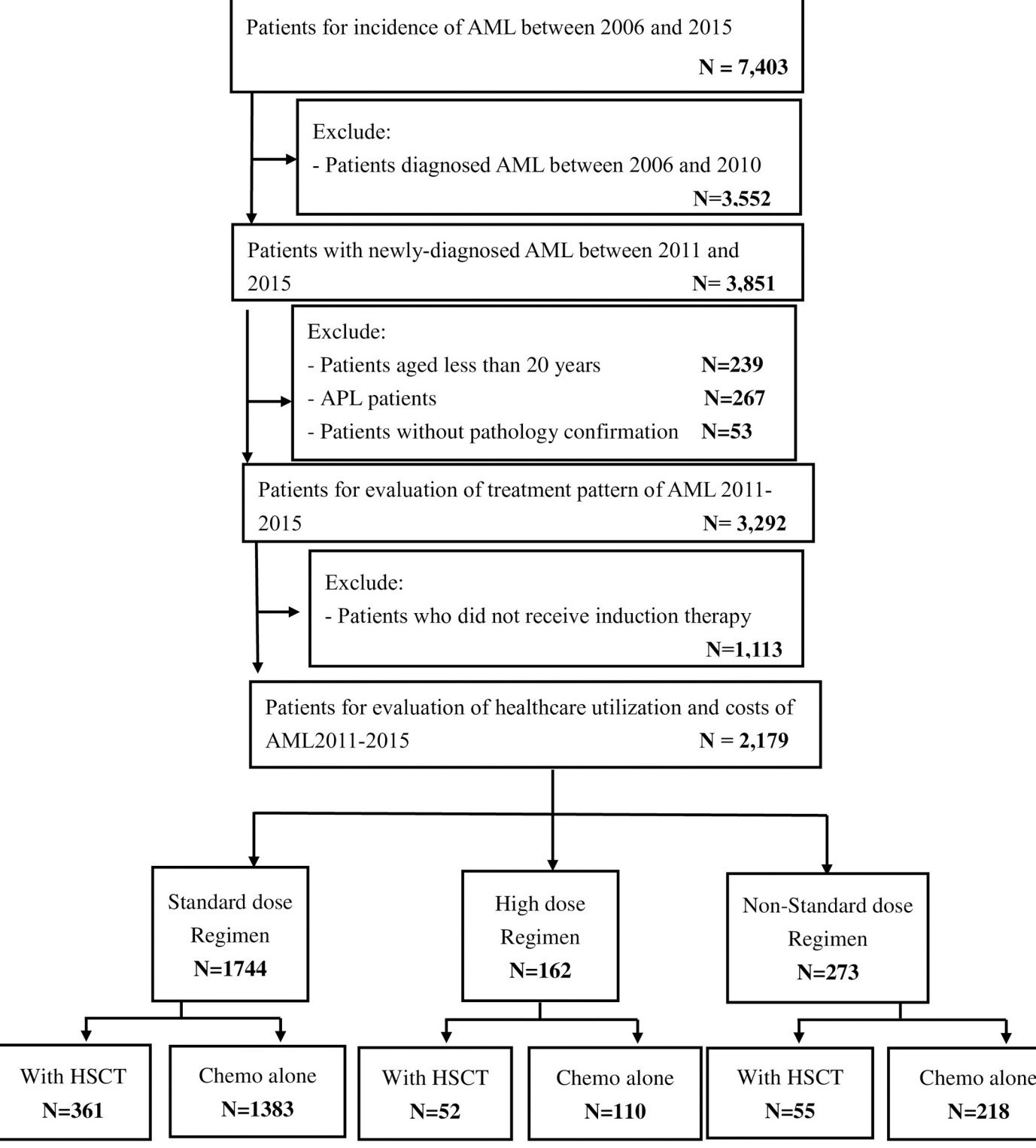

**Fig 1. Flowchart of the study cohort selection.** AML, acute myeloid leukaemia; APL, acute promyelocytic leukaemia; Chemo alone, chemotherapy alone; HSCT, hematopoietic stem cell transplantation.

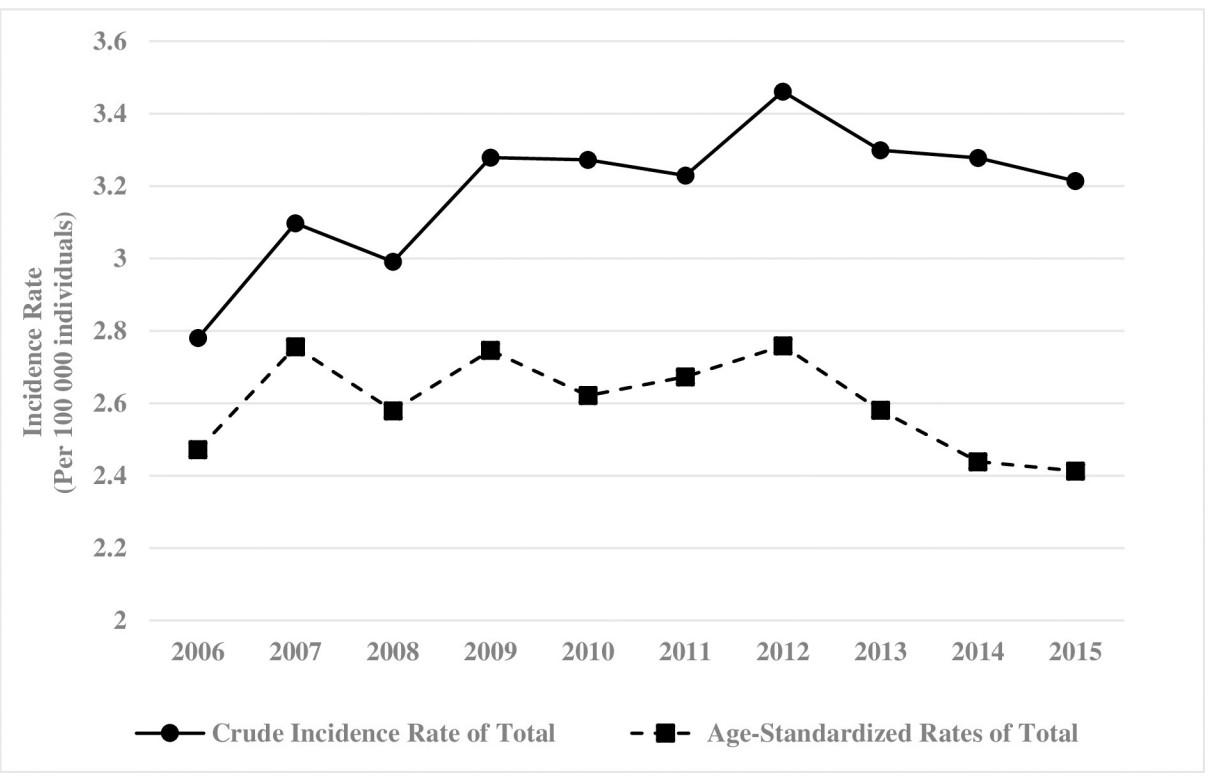

**Fig 2. 2006 to 2015 annual incidence and age-standardized rate of acute myeloid leukaemia.**

the second and the third year, number of hospitalization and the length of stay decreased in all regimens, but the median OPD visits in the patients who received N-SDAC with chemotherapy alone increased (19 and 29 in the 1st year and 2nd year, respectively; **S3 Table**).

Comparing the medical costs of chemotherapy alone and HSCT, the median medical costs per patient were the highest in the N-SDAC group with HSCT ($79,776 [IQR: $57,003-$121,595]). The medical costs of SDAC and HDAC group with HSCT were similar to the N-SDAC group with HSCT in the first year. (**Fig 4**). However, the median medical costs per patient were the highest in the HDAC group with chemotherapy alone ($42,271 [IQR; $26,751-$65,813]) and the lowest medical cost was in the SDAC group with chemotherapy alone ($36,199 [IQR; $19,518-$53,801]). The medical costs substantially declined in all groups in the second year, but the medical cost of the third year increased in the HDAC patients with HSCT and in the N-SDAC patients with HSCT (**Fig 4**).

## Discussion

To the best of our knowledge, this is the first population-based epidemiological study to investigate the incidence, treatment patterns, healthcare utilization and costs of AML patients in the Asian population. Similar epidemiological studies have been conducted and kept updated in the United States, Canada, Australia, the UK and Europe during the past decades [3,16–22]. There is only one population-based AML study from Taiwan reporting the AML incidence from 2000 to 2008 [23]. However, our study provides epidemiology of AML in recent years (2006–2015) and comprehensive analyses of treatment patterns, healthcare utilization and costs for AML patients that have not been done in other studies.

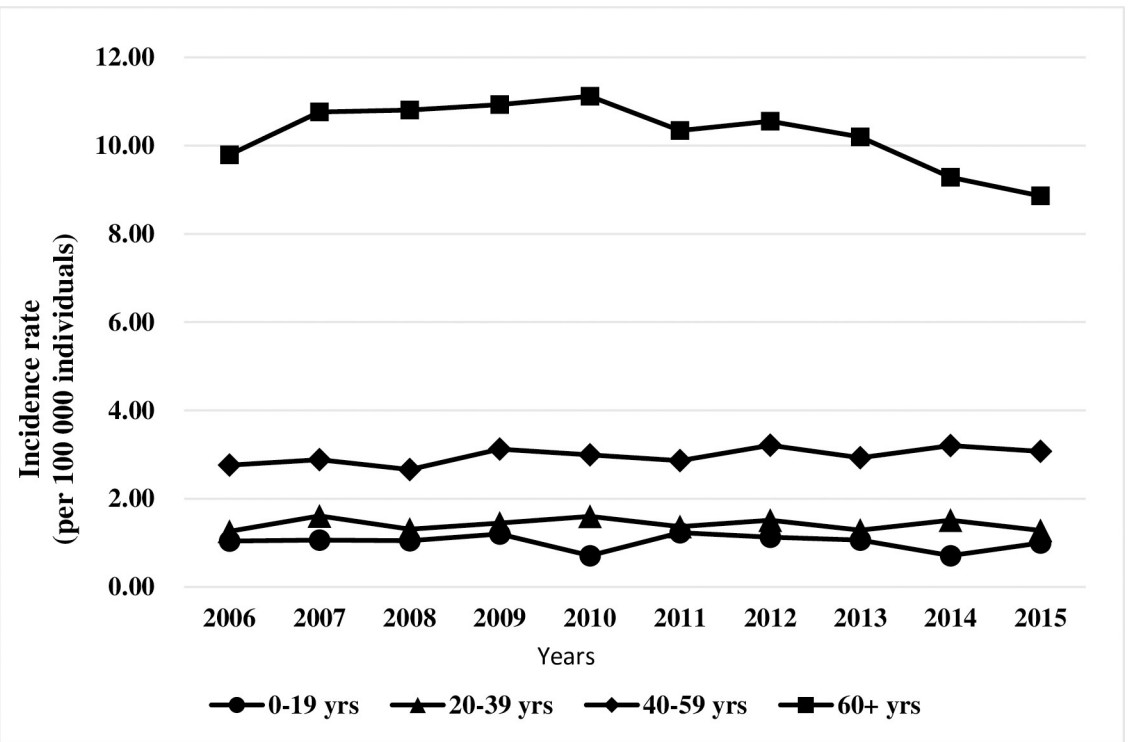

**Fig 3. 2006–2015 incidence of acute myeloid leukaemia (AML) in Taiwan, age-stratified.**

Our study illustrates several important points about Asian AML patients. First, the ASRs of AML ranged from 2.47 to 2.41 per 100,000 individuals during 2006 to 2015 in our study that was much lower than that in the UK (ASRs 4.05 per 100,000 individuals) [20], the US (ASRs 4.3 per 100,000 individuals) [24] and Canada (ASRs 3.47 per 100,000 individuals) [25]. The median age at diagnosis (57–61 years during 2006 to 2015) in Taiwan was younger than in the US (68 years) [24] and Spain (65 years) [26] but comparable to Canada (60 years) [25]. Comparing with Western countries, the lower incidence and younger age at diagnosis may be due to differences in the demographic structure of aging citizens in the population and ethnic characteristics.

Second, patients who aged less than 60 years at treatment (n = 1280) were more likely to receive SDAC (n = 1036, 80.90%) and HDAC (n = 108, 8.40%) regimen as induction therapy. However, the number of patients who aged 60 years and older (n = 899) decreased in SDAC group (n = 708, 78.8%) and HDAC group (n = 54, 6.0%). The treatment pattern of elderly patients (age ≥60) was more likely to receive N-SDAC regimen because the percentage of receiving N-SDAC regimen was 10.6% (n = 136) in age less than 60 years and increased to 15.2% (n = 137) in age 60 years and older. (**Table 1**) This study showed that treatment patterns of AML in Taiwan followed the NCCN guidelines [27] that AML patients who were aged less than 60 should be treated in SDAC or HDAC regimen. In contrast, patients aged 60 years and older should be treated with hypomethylating agent (HMA), which was defined as N-SDAC regimen in our study due to data limitation.

Third, our study provides insights regarding the healthcare utilization and costs of AML under a national health insurance program. Comparing with a previous study conducted by Preussler et al., which including AML patients aged 50 to 64 years old from 2007 to 2010 (221 with HSCT and 774 with chemotherapy alone) using MarketScan database in the United

**Table 1. Baseline and disease characteristics of AML patients who received induction therapy between 2011 and 2015.**

| | | Total (N = 2179) | | HDAC (N = 162) | | SDAC (N = 1744) | | N-SDAC (N = 273) | |
|---|---|---|---|---|---|---|---|---|---|
| | | n | (%) | n | (%) | n | (%) | n | (%) |
| Patient no. | | 2179 | (100) | 162 | (7.43) | 1744 | (80.04) | 273 | (12.53) |
| **Age at treatment, years** | | | | | | | | | |
| Median (IQR) | | 56 | (44–67) | 52 | (40–63) | 56 | (44–66) | 60 | (47–71) |
| 20–39 | | 406 | (18.63) | 39 | (24.07) | 329 | (18.86) | 38 | (13.92) |
| 40–59 | | 874 | (40.11) | 69 | (42.59) | 707 | (40.54) | 98 | (35.90) |
| 60–74 | | 647 | (29.69) | 43 | (26.54) | 515 | (29.53) | 89 | (32.60) |
| 75+ | | 252 | (11.56) | 11 | (6.79) | 193 | (11.07) | 48 | (17.58) |
| **Gender** | | | | | | | | | |
| | Female | 976 | (44.80) | 76 | (46.90) | 788 | (45.20) | 112 | (41.00) |
| | Male | 1203 | (55.20) | 86 | (53.10) | 956 | (54.80) | 161 | (59.00) |
| **Practice setting** | | | | | | | | | |
| | medical center | 1724 | (79.1) | 109 | (67.3) | 1397 | (80.1) | 218 | (79.9) |
| | others | 455 | (20.9) | 53 | (32.7) | 347 | (19.9) | 55 | (20.1) |
| **Charlson comorbidity Score** | | | | | | | | | |
| | 0 | 1192 | (54.7) | 107 | (66.0) | 964 | (55.3) | 121 | (44.3) |
| | 1–2 | 798 | (36.6) | 47 | (29.0) | 637 | (36.5) | 114 | (41.8) |
| | 2+ | 189 | (8.7) | 8 | (4.9) | 143 | (8.2) | 38 | (13.9) |
| **Time from diagnosis to induction therapy, days** | | | | | | | | | |
| Median (IQR) | | 7 | (3–18) | 5 | (3–10) | 7 | (3–16) | 13 | (4–37) |
| **Stem-cell transplant** | | 468 | (21.5) | 52 | (32.1) | 361 | (20.7) | 55 | (20.1) |
| **Time from induction therapy to HSCT, days** | | | | | | | | | |
| Median (IQR) | | 194 | (129–339) | 214 | (130–345) | 196 | (133–367) | 169 | (75–234) |
| **Follow-up time, months** | | | | | | | | | |
| | Mean (SD) | 20.6 | (22.7) | 21.3 | (21.9) | 20.8 | (22.8) | 18.9 | (22.2) |
| | Median (IQR) | 11.2 | (2.4–32.4) | 14.2 | (4.4–31.0) | 11.4 | (2.2–33.0) | 8.7 | (2.7–27.8) |

States, our number of OPD visits was lesser but the length of stay was longer. Preussler et al. reported that those with HSCT had a mean of 5.1 hospitalizations, 93.5 days of stay in hospital and 72.3 OPD visits, while those with chemotherapy alone has a median of 4 hospitalizations, 52.4 days of stay in hospital and 49.5 OPD visits [28]. Preussler et al. also reported the mean adjusted costs of patients with chemotherapy alone and with HSCT, which were $280,788 and $544,178 in the 1st follow-up year, respectively [28]. The median medical costs in our study for patients treated with chemotherapy alone was $42,271 for HDAC; $36,199 for SDAC; $36,250 for N-SDAC while patients treated with HSCT was $78,876 for HDAC; $78,593 for SDAC; $79,776 for N-SDAC. These discrepancies highlight the need to examine the healthcare utilization and costs for the different healthcare systems. However, the trends of healthcare utilization and costs in patients with chemotherapy alone and HSCT were comparable to Preussler et al that patients receiving HSCT had higher utilization and costs than patients receiving chemotherapy alone. (**Fig 4; S3 Table**)

## Limitations

There were several limitations in our study. First, as TCRD does not record body height and weight to do concise dose calculation for induction therapy, we stratified patients into SDAC, HDAC and N-SDAC regimens based on whether receiving cytarabine or not and the cumulative dose of cytarabine in the first cycle of chemotherapy. Second, because of a lack of

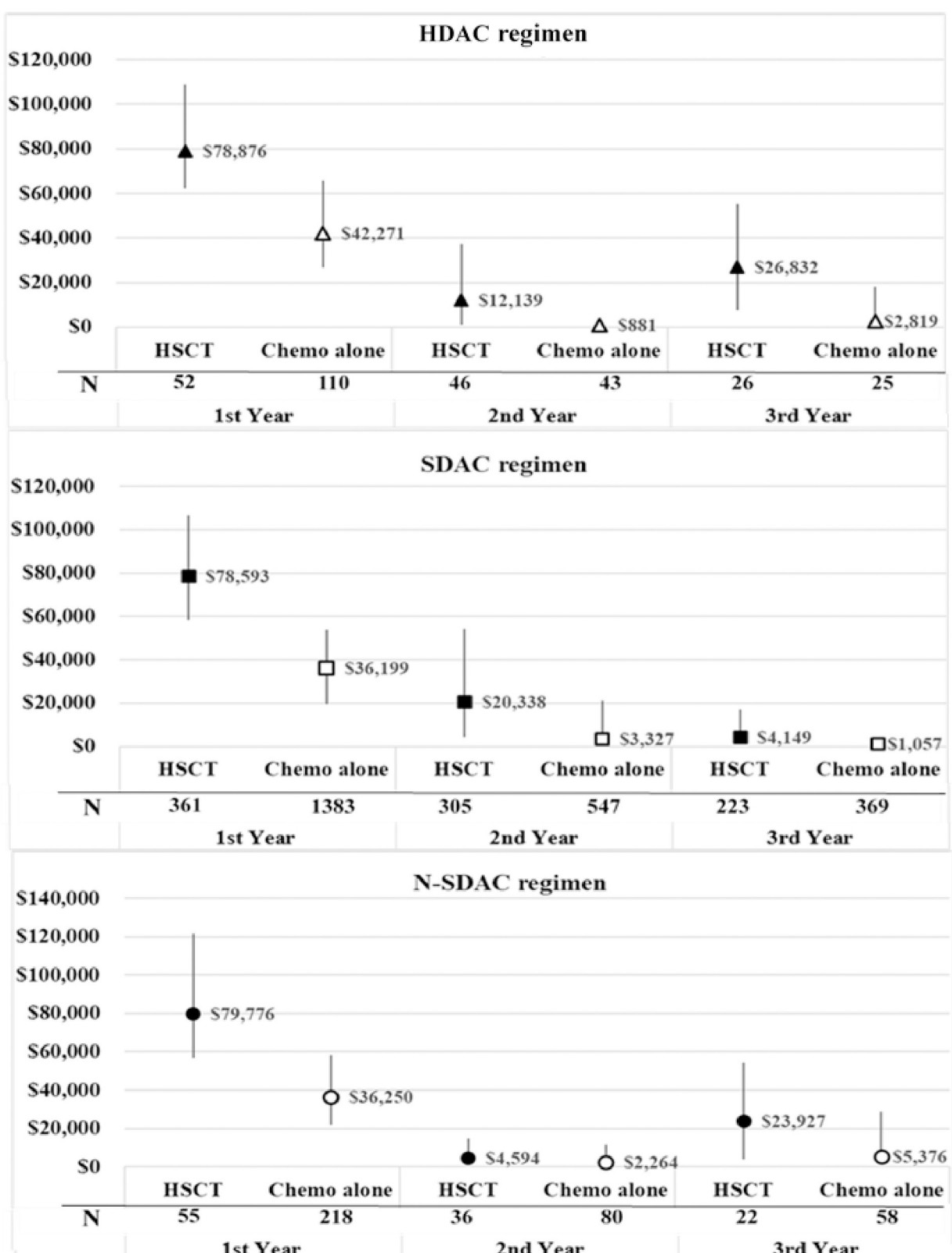

**Fig 4. Median (IQRs) medical cost per patient in the 1st, 2nd and 3rd year after AML diagnosis for patients who received chemotherapy alone or with HSCT, regimen stratify.**

information on cytogenetic and molecular analyses results in TCRD, the healthcare utilization and costs after receiving treatment may vary between those harboring different genetic abnormalities. Last, medications like azacitadine and sorafenib were not reimbursed by Taiwan's NHI; therefore, the costs of AML treatment could be underestimated in our study.

## Conclusion

Our study is the first Asian population-based study to provide updated and comprehensive information on the epidemiology and treatment patterns of AML patients. In addition, we report the healthcare utilization and costs of AML patients with chemotherapy alone and HSCT after the initial diagnosis of AML. These "real-world data" can fill up the knowledge gap and serve as good references for clinicians and policymakers to optimize AML management in the Asia Pacific region.

## Supporting information

**S1 Fig. Study design.**
(DOCX)

**S1 Table. 2008 WHO classification and ICD-O-3 codes.**
(DOCX)

**S2 Table. 2006–2015 annual incidence of AML.**
(DOCX)

**S3 Table. Healthcare utilization and costs of AML.**
(DOCX)

## Acknowledgments

We thank the National Health Insurance Administration (NHIA) and Health and Welfare Data Science Center (HWDC), Ministry of Health and Welfare, for making the databases used in this study available. However, the content of this article does not represent any official position of the NHIA or HWDC. The authors have full access to all of the data in the study and take responsibility for the integrity of the data and the accuracy of the data analysis.

## Author Contributions

**Conceptualization:** Huai-Hsuan Huang, Bor-Sheng Ko, Fei-Yuan Hsiao.

**Formal analysis:** Chen-Yu Wang, Ho-Min Chen, Fei-Yuan Hsiao.

**Methodology:** Huai-Hsuan Huang, Fei-Yuan Hsiao.

**Supervision:** Huai-Hsuan Huang, Bor-Sheng Ko, Fei-Yuan Hsiao.

**Writing – original draft:** Huai-Hsuan Huang, Chieh-Min Chen, Bor-Sheng Ko, Fei-Yuan Hsiao.

**Writing – review & editing:** Huai-Hsuan Huang, William Wei-Yuan Hsu, Bor-Sheng Ko, Fei-Yuan Hsiao.

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
