## [Decision Letter · Decision Letter 0]

19 Oct 2021

PONE-D-21-28087The Epidemiology, Treatment Patterns, Healthcare Utilizations and Costs of Acute Myeloid Leukaemia (AML) in Taiwan, 2006-2015PLOS ONE

Dear Dr. Hsiao

Thank you for submitting your manuscript to PLOS ONE. After careful consideration, we feel that it has merit but does not fully meet PLOS ONE’s publication criteria as it currently stands. Therefore, we invite you to submit a revised version of the manuscript that addresses the points raised during the review process.

We look forward to receiving your revised manuscript.

Kind regards,

Mohamed A Yassin, MD

Academic Editor

PLOS ONE

Journal Requirements:

“Huang HH, Chen CM, Wang CY, Hsu WWY, Chen HM, Ko BS and Hsiao FY reports grants from AbbVie Biopharmaceuticals , grants from Ministry of Science and Technology Taiwan ,  during the conduct of the study.

Hsiao FY, Huang HH and Ko BS contributed to the study concept and design. Chen HM, Wang CY and Hsiao FY acquired and analyzed the data. Huang HH and Ko BS interpreted the data. Huang HH, Chen CM, Hsiao FY and Ko BS drafted the manuscript. Hsiao FY, Hsu WWY, and Ko BS revised the manuscript. All authors read and approved the final manuscript. “

Additional Editor Comments:

Needs major revision not acceptable for publication in its current shape

Reviewers' comments:

Reviewer's Responses to Questions

**Comments to the Author**

1. Is the manuscript technically sound, and do the data support the conclusions?

Reviewer #1: Partly

Reviewer #2: Yes

Reviewer #3: Partly

2. Has the statistical analysis been performed appropriately and rigorously? 

Reviewer #1: Yes

Reviewer #2: Yes

Reviewer #3: Yes

3. Have the authors made all data underlying the findings in their manuscript fully available?

Reviewer #1: Yes

Reviewer #2: Yes

Reviewer #3: No

4. Is the manuscript presented in an intelligible fashion and written in standard English?

Reviewer #1: Yes

Reviewer #2: Yes

Reviewer #3: Yes

5. Review Comments to the Author

Reviewer #1: Overall, the study is nicely written but has a limited novelty and limited detailed analysis.

Specific comments:

- In the title, unless the period 2006-2015 is different from the usual, I suggest removing the period from the title.

- 36,250 is it repeated twice in the abstract

- the incidence rates reported in the introduction are old.

- In the study population section (methods), the year 2001 should be 2011.

- The sentence "We only analyze baseline characteristics of patients diagnosed with AML from 2011 to 2015 and received induction therapy within one year before the date of newly diagnosed AML" is confusing. Please, revise

- Receiving ara-c with anthracycline is a standard therapy while the definition used in the paper classifies this as non-standard. Please, clarify.

- Put the baseline characteristics as the first part of your results.

- replace Q1-Q3 with IQR

- Did the pattern of treatment change with age?

Reviewer #2: This is an interesting study and the authors have collected a unique dataset. The paper is generally well written and structured.

Only some minor comments:

- The authors need to check each plot in detail, they have to replace “Supplementary Table 4” with “Supplementary Table 3”.

- They have to remove commas (,) before “and”. For example, “healthcare utilization, and costs” (from Abstract: methods).

- In the Results section of the abstract there is a parenthesis (n=1744; ), where something is missing.

- Also in the same section syntactical error concerning the stated value of $36,250, it is repeated twice without making grammatical sense.

- In the same paragraph correct “remaing” with “remaining”, replace “costs was $78,876” with “costs were $78,876” and “For those received” with “For those who received”.

- In the Introduction part replace “100,000 persons” with “100,000 individuals”.

Reviewer #3: Treatment costs need to be itemized. Chemotherapy costs are elaborated upon. However it is not clear if supportive care expenses were also included or not. I think these should be included. Also hospital bed occupancy estimated cost and outpatient visit cost should also be computed.

The sentence "We only analyze baseline characteristics of patients diagnosed with AML from 2011 to 2015 and received induction therapy within one year before the date of newly diagnosed AML" is unclear.

NCCN abbreviation is mentioned in the abstract without adding its definition as : National Comprehensive Cancer Network

6. PLOS authors have the option to publish the peer review history of their article (what does this mean?). If published, this will include your full peer review and any attached files.

Reviewer #1: No

Reviewer #2: No

Reviewer #3: No

---

## [Author Response · Author response to Decision Letter 0]

25 Nov 2021

November 26, 2021

Dr. Mohamed A Yassin

Manuscript ID: PONE-D-21-28087

Title: The Epidemiology, Treatment Patterns, Healthcare Utilizations and Costs of Acute Myeloid Leukaemia (AML) in Taiwan.

Dear Dr. Yassin:

Thank you for the opportunity to respond to the reviewer’s comments regarding our manuscript. We extend our gratitude to the reviewers who provided constructive comments. More importantly, we appreciate the opportunity to respond to their reviews. Responding to the comments has allowed us to improve the overall quality of the work, which we hope has now addressed the issues raised. Below, you will find a point-by-point response to each of their concerns. We hope that you agree that the revised manuscript provides a novel contribution and is publishable in PLOS ONE.

Sincerely,

Fei-Yuan Hsiao, PhD

 

Reviewers' comments:

Reviewer #1: Overall, the study is nicely written but has a limited novelty and limited detailed analysis.

Comment 1: 

In the title, unless the period 2006-2015 is different from the usual, I suggest removing the period from the title.

[REPLY: Thank you for the comment. The title is revised as suggested.]

Comment 2: 

36,250 is it repeated twice in the abstract

[REPLY: Thank you for the comment. The correction is revised as suggested.]

Comment 3: 

The incidence rates reported in the introduction are old.

[REPLY: Thank you for the comment. We have added some new data in the incidence rates reported in the Introduction section. 

“In Europe, the incidence of AML has increased in years and the highest incidence was found in the UK, with an incidence rate of 4.05 cases per 100,000 individuals in 2017”.

To the best of our knowledge, these are the most updated incidence data we could retrieved from the literature.] 

Comment 4: 

In the study population section (methods), the year 2001 should be 2011.

[REPLY: Thank you for the comment. The correction is revised as suggested.]

Comment 5: 

The sentence "We only analyze baseline characteristics of patients diagnosed with AML from 2011 to 2015 and received induction therapy within one year before the date of newly diagnosed AML" is confusing. Please, revise

[REPLY: Thank you for the comment. For better clarity, the sentence is rephrased as “As the detailed information of induction treatment status was only available from 2011 in the NHIRD, we only analyze baseline characteristic of patients with newly diagnosed AML and received induction therapy from 2011 to 2015.”

Comment 6: 

Receiving ara-c with anthracycline is a standard therapy while the definition used in the paper classifies this as non-standard. Please, clarify.

[REPLY: Thank you for the comment. We agree that receiving cytarabine with anthracycline is a standard therapy; therefore, the definition of standard-dose cytarabine regimen in our study is patients who received an intravenous antineoplastic, immunomodulating agent (including anthracycline) and the accumulated dose of cytarabine is less than 3000 milligrams within 21 days from the diagnosis date of AML. 

The definition of a non-standard dose cytarabine regimen in our study is that patients who received an intravenous antineoplastic, immunomodulating agent and medications other than cytarabine, idarubicin and daunorubicin or received idarubicin or daunorubicin only within 21 days from the diagnosis date AML. 

To make the sentence clear, the sentence is rephrased as “All other patients who received medications apart from cytarabine, idarubicin and daunorubicin or received only idarubicin or daunorubicin within 21 days from the diagnosis date AML was defined as receiving a non-standard-dose cytarabine (N-SDAC) regimen.”

Comment 7: 

Put the baseline characteristics as the first part of your results.

[REPLY: Thank you for the comment.

As we reported epidemiology from 2006 to 2015 and then limited our study cohort as patients with newly diagnosed AML and received induction therapy from 2011 to 2015 for further analyses. We decided to kept the epidemiology as the first part of our results. However, we have modified the subtitles in the revised manuscript for better clarity.] 

Comment 8: 

Replace Q1-Q3 with IQR

[REPLY: Thank you for the comment. The modification is revised as suggested. In order to provide informative data, the value of IQR was presented as a range. 

Comment 9: 

Did the pattern of treatment change with age?

[REPLY: Thank you for the comment. 

Indeed, we found that age at treatment was different across different induction regimen received from our AML patients. The median age at treatment was the youngest for those received high-dose cytarabine (HDAC) regimens [52 years old] but was highest for those received non-standard-dose cytarabine (N-SDAC) regimen [60 years old]. (Table 1)]

 

Reviewer #2: This is an interesting study and the authors have collected a unique dataset. The paper is generally well written and structured.

Only some minor comments:

Comment 1: 

The authors need to check each plot in detail, they have to replace “Supplementary Table 4” with “Supplementary Table 3”.

[REPLY: Thank you for the comment. The correction is revised as suggested.]

Comment 2: 

They have to remove commas (,) before “and”. For example, “healthcare utilization, and costs” (from Abstract: methods).

[REPLY: Thank you for the comment. The correction is revised as suggested.]

Comment 3: 

In the Results section of the abstract there is a parenthesis (n=1744; ), where something is missing.

[REPLY: Thank you for pointing this out. The missing value is written as (n=1744; 80.04%) in the results section of the abstract.] 

Comment 4: 

Also, in the same section syntactical error concerning the stated value of $36,250, it is repeated twice without making grammatical sense.

[REPLY: Thank you for the comment. As mentioned in our reply to reviewer#1’s comment #2, the repeated value of $36,250 is deleted in the results section of the abstract.] 

Comment 5: 

In the same paragraph correct “remaing” with “remaining”, replace “costs was $78,876” with “costs were $78,876” and “For those received” with “For those who received”.

[REPLY: Thank you for the comment. The correction is revised as suggested.

Comment 6: 

In the Introduction part replace “100,000 persons” with “100,000 individuals”.

[REPLY: Thank you for the comment. The modification is revised as suggested.]

 

Reviewer #3: 

Comment 1:

Treatment costs need to be itemized. Chemotherapy costs are elaborated upon. 

[REPLY: Thank you very much for the comment. 

We understand the reviewer’s comment. Indeed, the chemotherapy costs are elaborated upon. However, we have adopted another approach, by categorizing AML patients into 6 groups according to their treatment regimens (high-dose cytarabine (HDAC), standard-dose cytarabine (SDAC) or non-standard-dose cytarabine (N-SDAC)) and status of HSCT. This approach allows us to capture the interaction between treatment regimens and HSCT on treatment costs of AML.

It is obvious that for patient who treated with chemotherapy alone, the treatment costs were highest for HDAC ($42,271) but comparable for SDAC and N-SDAC ($36,199, and $36,250). However, for those who received hematopoietic stem cell transplantation (HSCT) after induction therapy, their median medical costs were comparable to each other regardless which induction therapy they received ($78,876 for HDAC, $78,593 for SDAC, and $79,776 for N-SDAC). By adopting this approach, we also believe that chemotherapy costs were well characterized.] 

Comment 2:

However, it is not clear if supportive care expenses were also included or not. I think these should be included. 

[REPLY: Thank you very much for the comment. We would like to clarify with the reviewer that as the Taiwan’s National Health Insurance provide comprehensive coverage for healthcare services during hospitalization, supportive care expense are included in the calculation of the hospitalization cost. 

We have provided this information in the revised manuscript.]

Comment 3:

Also, hospital bed occupancy estimated cost and outpatient visit cost should also be computed.

[REPLY: Thank you for the comment. As mentioned in the reply to reviewer#3’s comment #2, we had computed the costs of hospitalization and the costs of OPD in Supplementary Table 2. 

Comment 4:

The sentence "We only analyze baseline characteristics of patients diagnosed with AML from 2011 to 2015 and received induction therapy within one year before the date of newly diagnosed AML" is unclear.

[REPLY: Thank you for the comment. As mentioned in the reply to reviewer#1’s comment #5. The sentence is rephrased.]

Comment 5:

NCCN abbreviation is mentioned in the abstract without adding its definition as: National Comprehensive Cancer Network

[REPLY: Thank you for the comment. The definition of NCCN is revised as suggested in the abstract.]

---

## [Editor Report · Decision Letter 1]

13 Dec 2021

The Epidemiology, Treatment Patterns, Healthcare Utilizations and Costs of Acute Myeloid Leukaemia (AML) in Taiwan

PONE-D-21-28087R1

Dear Dr.Fei

We’re pleased to inform you that your manuscript has been judged scientifically suitable for publication and will be formally accepted for publication once it meets all outstanding technical requirements.

Kind regards,

Mohamed A Yassin, MD

Academic Editor

PLOS ONE

---

## [Editor Report · Acceptance letter]

19 Dec 2021

PONE-D-21-28087R1 

The Epidemiology, Treatment Patterns, Healthcare Utilizations and Costs of Acute Myeloid Leukaemia (AML) in Taiwan 

Dear Dr. Hsiao:

I'm pleased to inform you that your manuscript has been deemed suitable for publication in PLOS ONE. Congratulations! Your manuscript is now with our production department. 

Kind regards, 

on behalf of

Dr. Mohamed A Yassin 

Academic Editor

PLOS ONE